# Combined Electrodialysis and Electrocoagulation as Treatment for Industrial Wastewater Containing Arsenic and Copper

**DOI:** 10.3390/membranes13030264

**Published:** 2023-02-23

**Authors:** Henrik K. Hansen, Claudia Gutiérrez, Jorge Leiva Gonzalez, Andrea Lazo, Marcela E. Hansen, Pamela Lazo, Lisbeth M. Ottosen, Rodrigo Ortiz

**Affiliations:** 1Departamento de Ingeniería Química y Ambiental, Universidad Técnica Federico Santa María, Valparaíso 2390123, Chile; 2Departamento de Ingeniería Química, Universidad de Santiago de Chile, Santiago de Chile 9170022, Chile; 3Instituto de Química y Bioquímica, Facultad de Ciencias, Universidad de Valparaíso, Valparaíso 2360102, Chile; 4DTU Sustain, Technical University of Denmark, 2800 Lyngby, Denmark; 5Escuela de Ingeniería Química, Pontificia Universidad Católica de Valparaíso, Valparaíso 2340025, Chile

**Keywords:** electrocoagulation, electrodialysis, copper and arsenic removal, current density

## Abstract

In copper smelting processes, acidic effluents are generated that contain inorganic contaminants such as arsenic and copper. Nowadays, the treatment of wastewater is done by physicochemical methods without copper recovery. Electrodialysis is an alternative process that can recover copper. Moreover, when electrocoagulation is applied to remove arsenic from wastewater, a more stable final sludge of less volume is obtained. The present research studies the application of a combined electrodialysis and electrocoagulation process to (1) recover Cu and (2) precipitate and remove arsenic simultaneously in the same batch reactor, using synthetic wastewater that simulates wastewater from a copper smelter. Copper and arsenic could be removed and separated by the electrodialysis part, and the electrocoagulation of arsenic was verified. With electrodialysis, the arsenic and copper removals were 67% and 100%, respectively, while 82% of the arsenic arriving at the electrocoagulation part of the cell could be precipitated and removed by this process. Initial concentrations were around 815 mg L^−1^ Cu and 7700 mg L^−1^ As. The optimal current was found to be 1.36 A due to the shorter treatment times necessary to get removal percentages, recovery percentages and energy/removed copper mass ratios in the same ranges as the values achieved with a current of 1.02 A. In summary, the combined process is a promising tool for simultaneous copper recovery and arsenic removal.

## 1. Introduction

Wastewater produced by mining activities often exceeds the limiting concentrations of different contaminants established by local authorities. For this reason, in Chile, a treatment of these effluents to lower the concentrations of the contaminants in order to fulfill the limits established by Chilean standard D.S. Nº90/00 is necessary [1].

Currently, the used treatment to reduce the concentration of inorganic contaminants in mining wastewater is based on the addition of chemical reagents, such as lime slurry, that facilitate the precipitation of the different metals by a pH increase of the wastewater, together with flocculant and coagulant addition. This procedure generates high economic costs due to the reagent consumption and the large volume of generated sludge, which requires further processing. Moreover, processes for the recovery of valuable metals are not implemented, and the waste is only disposed of with considerable contents of recoverable copper and other elements such as arsenic, zinc and lead [2,3,4].

Copper smelter wastewater contains several interesting elements that potentially could be recovered, for example, copper, cobalt, lead and tin [5,6]. Copper concentrations in these effluents are typically in the range of 350–750 mg L^−1^ [7,8]. Unfortunately, the presence of other elements, such as arsenic, makes copper recovery challenging. [9,10]. Normally, arsenic concentrations higher than 5000 mg L^−1^ can be found in untreated wastewater [7]. Furthermore, the predominant oxidation state of arsenic in the wastewater is (III) [11], and one main difference between copper and arsenic in these effluents is the charge of the ionic species found—copper is normally found as Cu^2+^, whereas arsenic (III) can be found either as negatively or un-charged species. This fact can incentivise the use of electrodialysis (ED) as a possible process to separate Cu from As since copper would migrate toward the cathode and arsenic eventually toward the anode. ED has been used widely in the concentration or desalination of aqueous media [12], and lately, it has been used for wastewater treatment in several types of industries [13]. 

ED has been tested with some success as a possible method to separate copper from other species, including arsenic, from synthetic wastewater [14]. In these cases, the focus was on obtaining a Cu-rich solution, whereas the behaviour of arsenic was not the main objective of the studies. Furthermore, the Cu-to-As mass ratios in the solutions studied were around 3, where this ratio in real copper smelter wastewaters ranges between 0.05–0.2 [7]. 

One concern with the suggested process is the final use of the separated arsenic. Since arsenic does not have a great recoverable value, it ends up typically immobilised in solid waste material. Typically, in the treatment of copper smelter wastewater, large amounts of arsenic-containing sludge are produced, so if the volume of this solid waste could be reduced, then it would be favourable. Electrocoagulation (EC) is a process that has previously shown the potential to reduce the volume of the final waste [15]. EC is by now a well-known wastewater treatment method based on the use of sacrificial anodes that would be the source of either iron or aluminium for the precipitation/coagulation process, and it has been used in the treatment of several types of wastewater [16]. In the actual case, arsenic is precipitated with ferric ions produced by the anodic reaction, generating a very stable precipitate [17].

The aim of this research is to study a new technology that combines ED and EC to remove the inorganic contaminants arsenic and copper present in wastewater from a copper smelting process, applying electrochemical processes at a low voltage. ED would separate the positively charged copper from the neutral or negatively charged arsenic, even if the concentration of arsenic is much higher than copper. EC would precipitate arsenic in the same cell in the anodic compartment using the same electric current as for the ED part to produce a much lesser volume of sludge with arsenic compared to conventional arsenic-treating precipitation processes. The reagent for the precipitation (in this case, ferric ions) would be generated in situ at the anode. Afterwards, the treated wastewater could be discharged or reused in the copper smelter plant. Therefore, several advantages of the suggested process can be listed: (1) recovery of copper, (2) less final As-sludge, (3) treated wastewater, and (4) less reagent addition. The efficiency of the process is evaluated in a four-compartmental experimental batch setup, where the copper removal and the arsenic precipitation are monitored, and the studied variables are treatment time and current density. 

## 2. Theoretical Background

ED is a conventional technique used to transport, separate and concentrate ions by means of an electrical field [18]. In this process, the added electrical energy is converted to the movement or migration of ions according to their electric charge across a system of ion exchange membranes [19]. When an electrical field perpendicular to the membranes is applied, anions will be transported to the anode and the cations to the cathode [3,20,21].

On the other hand, EC is a process used to remove suspended or dissolved contaminants from aqueous solutions [16,17]. This technique is based on the application of an electric current between electrodes submerged in the solution to be treated. Among the most used materials for electrodes are iron and aluminium. In the case of iron electrodes, the applied current oxidises and dissolves the anode, and ferrous ions in solution are then oxidised by oxygen or another oxidant, producing hydrous ferric oxides (HFO) that form flocs on which arsenic and other contaminants are adsorbed and/or co-precipitated. It is possible to remove these flocs from the solution by a separation method, obtaining a clean solution [2,4,22,23]. The main factors affecting EC are (i) surface area of electrodes, (ii) types of electrodes, (iii) current density, (iv) pH of the solution, (v) oxidant, and (vi) mixing in the coagulation process.

It is important to highlight that when the current density is too high, ferrous ions are produced at higher rates causing passivation of the anode. The passivation of the anode takes place when the concentration of iron in the solution near the electrode is too high, thus producing a superficial layer of oxides on the electrode [24]. A sign of this is the increase in the cell’s electrical resistance due to the lack of circulating electrons. To avoid the passivation of electrodes, electrode polarity inversion is commonly used, but this option is not possible in this case since the electric field should be in the same direction always. The aeration employing air bubbling produces an increase in the turbulence inside the cell, improving the mass transfer and giving the necessary oxygen to oxidise Fe^+2^ to Fe^+3^ and producing ferric hydroxides [17,25]. The pH directly affects the formation of hydroxides, and the formation is lower with a pH under 5 [17,26]. Therefore, it is important to maintain a pH over 5 in the section where the EC is carried out.

The two main electrode reactions in the reactor should be:

At the cathode (Reduction):

Water electrolysis is occurring:(1)$$2H_{2}O+2e^{-}\to 2OH^{-}+H_{2}↑$$

At the anode (Oxidation):

When an electric current is applied, the iron (Fe) electrode is oxidised. This process occurs according to the following reaction:(2)$$Fe_{\left(s\right)}\;\to \;Fe_{\left(ac\right)}{}^{2+}+2e^{-}$$

Reaction (2) represents the release of ferrous cations to aqueous media. 

Reactions (3) and (4) show the oxidation of Fe^+2^ to Fe^+3^ due to the oxygen addition to the solution through air bubbling. 

In an acidic medium:(3)$$4Fe_{\left(ac\right)}{}^{2+}+O_{2\left(g\right)}+4H_{3}O^{+}\;\;\;\to \;\;\;4Fe_{\left(ac\right)}{}^{3+}+6H_{2}O$$

In basic medium:(4)$$4Fe_{\left(ac\right)}{}^{2+}+O_{2\left(g\right)}+2H_{2}O\to 4Fe_{\left(ac\right)}{}^{3+}+4\;OH^{-}$$

Reactions (3) and (4) are necessary to achieve the removal of arsenic.

Iron hydroxide formation is represented by the following reactions:(5)$$Fe^{3+}+3OH^{-}\;\;\;\to \;\;\;\;\;\;Fe{\left(OH\right)}_{3}↓$$
(6)$$Fe^{3+}+3H_{2}O\;\;\;\to \;\;\;\;\;\;Fe{\left(OH\right)}_{3}↓+3H^{+}$$

At pH 2.0, arsenic is present in solution as arsenic acid (H_3_AsO_4_ or H_2_AsO_4_^−^) and arsenous acid (H_3_AsO_3_ or H_2_AsO_3_^−^) depending on the oxidation level of arsenic and redox potential. If As(III) is present in the aqueous solution, it will be oxidised to As(V) by the oxygen in the air. 

The arsenic adsorption by iron hydroxides corresponds to the following: (7)$$\alpha Fe{\left(OH\right)}_{3}+\beta As\;\;\to \;\;\;\;\;\;\alpha Fe{\left(OH\right)}_{3}\times \beta As$$

Moreover, As(V) in solution reacts with ferric ions to produce ferric arsenate by the reaction:(8)$$Fe_{\left(ac\right)}{}^{3+}+AsO_{4}{}^{3-}\;\;\;\;\to \;\;\;\;FeAsO_{4}↓$$

## 3. Experimental

The rectangular combined ED and EC cell is shown in Figure 1, and its main feature is to allow simultaneously both the ED and EC processes to be carried out batch-wise. The cell is formed by four sections, and two electrodes are used, one of stainless steel (cathode) placed in Section 1 and another of carbon steel (anode) placed in Section 4, where ferrous ions are generated, thus producing the EC. The purpose of the reactor setup is to achieve the separation of Cu and As from the wastewater (placed in Section 3), first by the transportation of Cu and As to Section 2 and Section 4, respectively, and secondly by precipitating As with EC in Section 4.

The working cell has four sections, each of them with a different function:

Section 1: In this section, a rectangular homemade stainless steel cathode with an area of 0.0025 m^2^ is placed. The section contains a solution of 0.5 M sulphuric acid, and an anion exchange membrane is placed adjacent to Section 2 to avoid the flow of cations to the cathode from that section.

Section 2: In this section, copper recovered from the wastewater loaded in Section 3 should be accumulated. This is due to the use of a cation exchange membrane between Section 2 and Section 3 that allows the cations (in this case, preferably Cu^2+^ due to the conditions of the wastewater) to flow in the direction of the electric field. Initially, this section will be loaded with 0.5 M sulphuric acid. 

Section 3: In this section, wastewater to be treated will be loaded, with a pH initially adjusted to 2.0, according to results found from [3], where this pH level was found to be adequate for ED transport of Cu^2+^.

Section 4: In this section, a rectangular homemade carbon steel anode with an area of 0.0066 m^2^ is placed. The anode will be oxidised in the process producing ferrous ions, which will be used later in the EC process. This section also contains a solution of 4 g L^−1^ NaCl to increase the electrical conductivity, and 5% NaOH is used to adjust the solution pH to 6.5. In this section, anions from the loaded solution of Section 3 are accumulated, and the EC of arsenic is obtained. As(III) flows as $H_{2}AsO_{3}{}^{-}$ or $HAsO_{3}{}^{2-}$ from Section 3 to Section 4 due to the direction of the electric field and the anion exchange membrane between the sections. An aeration system formed by a glass ring connected to a compressor was used to oxidise Fe^2+^ to Fe^3+^ (and eventually As(III) to As(V)) and to maintain the stirring.

The internal width and height of the cell were 9 and 10 cm, respectively. The cell compartments were open at the top but covered by a plastic film to reduce evaporation. The solution volume loaded in each section was 450 mL for experiments in Series 1 and 400 mL for experiments in Series 2. The membranes were from Membranes International Inc (Ringwood, NJ, USA), a cation exchange membrane CMI-7000 and an anion exchange membrane AMI-7001. CMI-7000 is a strong acid cation exchange membrane with a sulphonic acid functional group. AMI-7001 is a strong base anion exchange membrane with the functional group of quaternary ammonium. Both membranes have a polystyrene structure cross-linked with divinylbenzene. The current density for ion transport across the cell was calculated based on the area of the ion exchange membranes that was covered by the solution during experiments. The area was found to be around 0.007 m^2^. 

The DC-power supply was a model 382285 from Extech Instruments (Waltham, NH, USA). Two multimeters, Uni-Trend International Limited (Dongguan, China) model UT60 Series, were used for electric current and cell voltage measurements during experiments. A pH meter, model Orion 370 (Thermo Scientific, Beverly, MA, USA), was used to adjust the solution pH. To filter the samples, a vacuum pump model WOB-L 2522 from Welch, Ilmenau, Germany), Büchner funnel, Kitasato flask and filter paper grade Nº131 (Advantec, Dublin, CA, USA) were used. Samples were analysed by the Chilean Official Standard method NCh 2313/10 of 96. An Atomic Absorption Spectrophotometer Varian model SpectrAA 55 (Palo Alto, CA, USA) was used. 

The synthetic wastewater was prepared by dissolving 3.158 g L^−1^ of CuSO_4_·5H_2_O and 14.047 g L^−1^ of NaAsO_2_ (both analytical grade) in distilled water. 97.9% sulphuric acid (J.T. Baker, Phillipsburg, NJ, USA) or 0.5 M NaOH prepared by dissolving 98% extra pure sodium hydroxide pellets (Loba Chemie, Mumbai, India) were used to adjust the pH of the solutions.

Table 1 shows the experimental details, operational parameters and obtained results for the experiments. Series 1 was conducted with a current of 1.36 A and contained four experiments with a maximum duration of two hours, while Series 2 included three experiments carried out with a current of 1.02 A and a maximum duration of 2.45 h. The current densities were selected based on previous experimental results, where only the ED part of the process was studied [3]. Samples were taken from Section 2, Section 3 and Section 4 at the end of each experiment. Moreover, the current and the cell voltage were measured. In experiment 1, the As and Cu concentrations were measured in Section 1 and in the different membranes too, but since all the concentrations were lower than 0.1 mg L^−1^, these measurements were not done in the remaining experiments, and their contributions to the mass balance were neglected.

## 4. Results and Discussion

The experimental results are shown in Table 1, where the arsenic and copper concentrations in Section 2, Section 3 and Section 4 over time for the two series can be found. Experiments 1 and 5 were carried out for a sufficiently long time in order to deplete Section 3 of ions so that no current could be conducted across the cell. It was possible to observe an increase in copper and arsenic removal from Section 3 as a function of treatment time. For both current densities, a copper removal of 100% was achieved for treatment times over 2 h. In the case of arsenic removal, only values close to 67% were achieved in both cases.

To analyse the arsenic and copper removal from Section 3, where initially the synthetic wastewater was added, the following equation was used:(9)$$\;\;\;\;\;\;\;\;\;\;\;\;rem_{i.j,3}=\frac{M_{initial\;i.j,3}-M_{final\;i.j,3}}{M_{initial\;i\;.j,3}}\times 100\;\%$$
where *rem_i,j,_*_3_ is the removed mass percentage from Section 3 of element *j* (copper or arsenic) in experiment *i*. *M_initial i,j,_*_3_ is the initial mass in Section 3 of element *j* in experiment *i*. Finally, *M_final i.j,_*_3_ corresponds to the final mass in Section 3 for element *j* and experiment *i*.

To analyse the arsenic and copper mass percentage content in Section 2 and Section 4, the following relation was used:(10)$$\;\;\;\;\;\;\;\;\;\;\;\;rec_{i.j,\;k}=\frac{M_{final\;i.j,k}}{M_{initial\;i,j,3}}\times 100\;\%$$
where *rec_i,j,k_* corresponds to the mass percentage of element *j*, in the solution of Section *k*, in experiment *i*. *M_final i.j,k_* is the final mass of element *j* in the solution of Section *k* and experiment *i*.

### 4.1. Arsenic Behaviour

In Figure 2, the removal percentage of arsenic from Section 3 over time is presented. For the two current densities, the increase in treatment time produces an increase in arsenic removal showing a linear relation. When the maximum arsenic removal percentages are compared, 67.3% removal after 2 h of treatment was obtained with 1.36 A, while with 1.02 A, 66.4% As removal was obtained after 2.45 h. These values are quite similar, but this could also be explained by the amount of charge passed through the system in each case, which were also similar: 7452 C and 7103 C, respectively.

To estimate the necessary time to reduce the concentration to 1 mg As L^−1^, which is the maximum level for discharge to the environment according to Chilean authorities [1], the method of least squares was used to extrapolate the removal, assuming that the arsenic transport rate is constant. Thus, a time of 3.3 h for 1.36 A and 4.0 h for 1.02 A was obtained. Furthermore, Section 2, where copper is concentrated during the process, should preferably not have a high content of arsenic at the end of experiments if copper must be recovered.

Section 4 has the function of receiving and process as much arsenic as possible. Arsenic anions migrate to this section by the electric field after dissociation of arsenous acid (As(III)) and arsenic acid (As(V)), depending on the pH of the solution and redox potential, and according to the Reactions (11)–(14) [27].
(11)$$H_{3}AsO_{3}+H_{2}O\underset{}{\Leftrightarrow }H_{2}AsO_{3}^{-}+H_{3}O^{+}\;\;\;\;\;pK_{a}^{1}=9.29$$
(12)$$H_{3}AsO_{4}+H_{2}O\underset{}{\Leftrightarrow }H_{2}AsO_{4}^{-}+H_{3}O^{+}\;\;\;\;\;pK_{a}^{1}=2.26$$
(13)$$H_{2}AsO_{4}^{-}+H_{2}O\underset{}{\Leftrightarrow }HAsO_{4}^{2-}+H_{3}O^{+}\;\;\;\;\;pK_{a}^{2}=6.76$$
(14)$$HAsO_{4}^{2-}+H_{2}O\underset{}{\Leftrightarrow }AsO_{4}^{3-}+H_{3}O^{+}\;\;\;\;\;pK_{a}^{3}=11.29$$

Moreover, in Section 4, the EC of arsenic is carried out. The arsenic content in the precipitated solid was not measured but estimated. To estimate the amount of coagulated arsenic due to EC, an arsenic mass balance for each experiment was made. The following relationship must be fulfilled:(15)$$M_{EC\;i,As,4}=M_{initial\;i,\;As,\;3}-M_{final\;i,\;As,3}-M_{final\;i,As,2}-M_{final\;i,As,\;4}$$
where $M_{EC\;i,As,4}$ is the mass of electrocoagulated arsenic in Section 4 for experiment *i*. *M _initial i,As,3_* is the initial mass of arsenic in Section 3, for experiment *i*. *M _final i,As,3_* is the final mass of arsenic in Section 3, for experiment *i*. *M _final i,As,2_* is the final mass of arsenic in Section 2, for experiment *i*. *M _final i,As,4_* is the final mass of arsenic in the solution of Section 4 for experiment *i*. As mentioned before, no arsenic was found to be present in Section 1 or the membranes, so these parts are neglected in the mass balance.

According to that, the total percentage of removed arsenic by coagulation is expressed by Equation (16).
(16)$$\;\;\;\;\;\;\;\;\;\;\;\;\%As_{EC\;total}=\frac{M_{EC\;i,As,4}}{M_{initial\;i,\;As,\;3}}\times 100\;\%$$

The tendency that the amount of coagulated arsenic is increasing over time can be seen in Table 1. The percentage of coagulated arsenic compared to the initial arsenic content in Section 3 is not higher than 55% and is very similar in both series at the end of the experiments when the electric current in the system approaches 0. Still, only a little arsenic is transported to Section 2, and it could be expected that more processing time could have removed more arsenic from Section 3 to Section 4. However, since the electrical conductivity of the system is very low after, for example, 2 h at 182 A m^−2^, it is not recommended to continue further in time. Probably Section 3 is depleted of most ions, and the remaining arsenic would be uncharged species. Therefore, further research should be focussed on the determination of arsenic speciation in this section. A suggestion could be that an external EC unit could treat the solution leaving Section 3 and remove the remaining arsenic, which could be possible since all copper has left this section. Another way to increase the As removal could be with pH control in Section 3, assuring that As always is a negatively charged species. Oxidation of As(III) to As(V) in Section 3—for example, with air or oxygen—could also improve arsenic transport to Section 4 since H_2_AsO_4_^−^ is expected to be more mobile than the As(III) species, also in ED processes [28]. In addition, As(III) is considered a more toxic species than As(V) [29], so it would be favourable from an environmental point of view to oxidise arsenic. Anyway, for practical purposes, the process should be developed as a continuous process since it is not desirable to reach the point where zero current is passed across the cell.

An arsenic mass balance for Section 4 is shown in Table 2, including the total incoming As (arsenic found in the solution and precipitate) and the coagulated As. A maximum percentage of coagulated arsenic equal to 82% is observed for experiments 1 and 5, which indicates that the oxidation reactions and adsorption to HFOs are not fast enough to precipitate all arsenic during the time of treatment. The oxidation could be improved with a higher concentration of an oxidant (for example, pure oxygen) in Section 4, the use of a vigorous stirring of the solution, or by hydrogen peroxide addition, which has been tested previously in EC [2,4]. This would decrease the oxidation time and increase the oxidant mass transfer in the section.

Figure 3 shows the percentage of arsenic (compared to initial mass) in Section 3, in the solution of Section 4 and the calculated amount in the EC precipitate in Section 4 for the two experimental series. The amount in Section 3 decreases almost linearly, whereas the amount in the solution of Section 4 increases until around 1 h of treatment, then the amount tends to reach a constant value, while the percentage of coagulated arsenic increases after that point. However, the amount of Fe^2+^/Fe^3+^ in Section 4 is, of course, directly related to the charge passed through the system since it is expected that the main anode reaction is Fe → Fe^2+^ + 2 e^−^ [30]. Therefore, the amount of HFOs that would be generated will increase with time [31]. So, it is possible to conclude that a high generation of Fe^2+^ ions, which are oxidised further to Fe^3+^, produces the greatest quantity of adsorbed arsenic on the HFOs.

### 4.2. Copper Behaviour

Table 1 includes the Cu concentrations in Section 2, Section 3 and Section 4 and the Cu removal for the different experiments. As mentioned before, most of the Cu is accumulated in Section 2 with time. This is shown in Figure 4, which represents the copper concentration in Section 2 as a function of time. An increase in copper concentration in series 1 and 2 is observed, and, in both cases, it is not possible to reach the copper amount initially present in Section 3. In this context, it is important to note that over 90% of the initial copper in the wastewater was found in Section 2 at the end, which implies that the recovery percentage is significant.

It could be interesting to see if the copper removal is dependent on the passed electric charge when the treatment is applied, and this is illustrated in Figure 5, which shows the copper removal percentage from Section 3 for the two experimental series as a function of the charge. The two curves overlap quite clearly, indicating that the passed electric charge is more process-determining than the treatment time itself. Furthermore, the curves also seem to have a linear behaviour. In both experimental series, the maximum removal was approximately 100%, reaching concentrations lower than 1 mg L^−1^ for the same amount of passed charge but at different times. With a current of 1.36 A, this was achieved after 2 h, whereas for 1.02 A, 2.45 h was needed. It seems that around 7100–7400 C are needed to deplete Section 3 totally of ions for the experimental conditions chosen in this work.

In Section 4, where most of the arsenic was accumulated, the concentration of copper was lower than 1 mg L^−1^ in the solution in both series. From the mass balance, it is possible to notice a small lack of copper that is lower than 5 % compared to the initial mass. Some copper could have reached Section 4, and coagulated/precipitated there after crossing the anion exchange membrane by diffusion, but the amount is uncertain because the copper content in the precipitate was not measured. In any case, the direction of the electric field is opposite and disfavouring this copper transport.

It is noted from Table 1 that for both current densities, the concentrations of As were lower than 80 mg L^−1^ in Section 2, meaning that less than 1% of the total mass of arsenic originally present in Section 3 at the start of each experiment is ending in Section 2. Here the movement of arsenic to Section 2 was due to diffusion phenomena and the permeability of the membrane. The final Cu-to-As mass ratio in Section 2 was around 10, whereas this ratio was lower than 0.05 in Section 3 when the experiments started. Therefore, the separation of Cu from As and copper concentration was found to be successful, and a second treatment unit could process and separate the elements further if necessary.

### 4.3. Current Efficiency

It is of crucial importance for the process efficiency to know how much of the applied current is used to transport the element of interest across the different membranes. This current efficiency with respect to the transport of a specific element is based on the relation between the mass of an element effectively transported and the theoretical (maximum) mass of the element to be transported, according to the intensity of the applied current. The maximum mass of one element to be transported corresponds to when all the direct current is used only for the transport of this element as ionic species. In this case, current efficiency would be calculated as the removed element from Section 3.

The relation between the experimental mass removed from element i, *M_expi_,* and the theoretical mass to be removed from element i, *M_theoi_,* is:(17)$$CE\left(\%\right)=\frac{M_{expi}}{M_{theoi}}\times 100$$

The calculations of the theoretical mass transported are based on the equations used to estimate theoretical electrochemically deposited metals by applying Faraday´s law. So, it is possible to calculate the theoretical mass removed from the electrolyte and the theoretical mass is expressed as follows: (18)$$M_{theoi}=\frac{I\times \Delta T\times PA_{i}}{F\times n}$$
where *I* is the intensity of the applied current, A; Δ*T* is the application time of the current, s; *PA_i_* is the atomic weight of element *i*, g mol^−1^; *n* is the number of electrons by mol of element *i* (corresponds in this case to the valency of the ionic species that is transported); and *F* the Faraday constant with a value of 96,485 C mol^−1^.

Using Equation (17), the current efficiency is obtained for copper and arsenic, and the results are shown in Table 3. In these calculations, it is not the electrode reactions that are important but the ionic species that are crossing the ion exchange membranes on both sides of Section 3. Copper is assumed to be transported as Cu^2+^, whereas arsenic could be transported as monovalent anions such as H_2_AsO_3_^-^. In the case of arsenic, this could be rather approximative since the speciation is not determined, but due to the initial and final pH in Section 3 (around 2 and 7, respectively), this speciation of As(III) is the most dominant anionic species [32].

The obtained current efficiencies for copper do not exceed 20%, but the values are higher than those obtained in previous works, where the calculated current efficiency was lower than 5% [3]. Arsenic current efficiencies are somewhat higher, but probably some of the arsenic is also present as uncharged species (H_3_AsO_3_) or as divalent anions (HAsO_3_^2−^), and this could influence the calculations. One recommendation could be to oxidise arsenic to As(V) in the wastewater before adding it to Section 3 since arsenic in oxidation state (V) is known to be more mobile in aqueous solutions [33], and higher amounts of arsenic would be anionic species (e.g., H_2_AsO_4_^−^) in the pH range used and in an oxidised media [29]. From Table 3, it looks like Cu is carrying the current better at the beginning of the experiments, whereas later, the current efficiency with respect to As increases—maybe because now the Cu concentration in Section 3 is lower. It could be interesting to analyse if the selectivity of the different membranes has importance for the process. Ion exchange membrane selectivity has great importance for various ionic separation processes [34]. In this work, the main elements in the prepared solutions were copper and arsenic, and therefore it is necessary to carry out further research on real copper smelter wastewater to see if the tendencies are the same.

### 4.4. Behaviour of Voltage and Current

The experiments were carried out at a constant current. This means that the cell voltage varies during the process. The current was maintained constant over time until Section 3 became depleted of ions. At that point, the maximum voltage of the power supply was reached, and the current began to decline since the cell resistance increased due to the low concentration of ions to be transported by the electric field.

Figure 6 and Figure 7 show the behaviour of current and voltage over time, respectively. From Figure 6, it is seen that the decrease in current started after 1.3 h for 1.36 A; meanwhile, for 1.02 A, this decrease was initiated at 1.6 h. Figure 7 shows the evolution of cell voltage over time until it reached the maximum capacity of the power supply, which is approximately 60 V. This happens because the system reaches a maximum value of voltage when a lower quantity of ions is present (in Section 3), and this means an increase in the resistance. When the experiment does not reach the maximum value of the voltage, not all ions have been transferred in the system, which means that ions are available to be transferred between the different sections. Therefore, for practical purposes, this total depletion point should not be reached in an actual treatment plant. Since these experiments were carried out batch-wise, this could indicate that treatment should be done in a continuous way, with a residence time lower than the time needed for the total depletion of ions from Section 3. From Figure 6 and Figure 7, it can also be seen that the reproducibility of the experiments is quite good since the curves for the different experiments overlap each other—at least considering current and voltage.

Figure 8 shows the electrical resistance across the whole cell over time for the two series of experiments, at 1.36 A and 1.02 A. From Figure 8, it is clearly seen that the cell electrical resistance is constant at a low level until the ion strength of the solution in Section 3 gets lower, and thereafter the resistance increases rapidly. This significant increase in electrical resistance after Section 3 is depleted of ions; therefore, it does not seem to be the result of an increase of the real membrane resistance but is caused by the diffusion boundary layer resistance, which is more pronounced at lower solution concentrations [35,36]. Furthermore, the limiting current for the membranes has been overpassed, meaning that water splitting occurs, which also adds to cell resistance [37]. In real copper smelter wastewater, it would also be interesting to see if this ion depletion occurs or if most copper and arsenic could be removed before reaching this point in preference to other ionic species.

### 4.5. Power and Energy Consumption

Power and energy are used to compare the efficiency of the process in relation to energy consumption since both parameters are related to the operational costs of the system.

The following relation estimates the consumed power by the system:(19)P=i×V
where *P* is the consumed power by the system in W, *i* is the applied current in the cell in A, and *V* is the applied voltage in the system in *V*.

To determine the consumed power, experiments 1 and 5 were used, which according to Table 1, corresponds to the experiments where the highest removals of copper for the different applied currents were obtained. Figure 9 shows the evolution of the power over time for the two studied currents.

The consumed energy to obtain 100% of copper removal was determined by the following relation:(20)$$\;\;\;\;\;E=\int_{0}^{t}V\times i\;dt$$
where *E* is the consumed energy to remove 100% of copper in kJ and *t* is the time in s.

Therefore, the consumed energy would correspond to the area underneath each curve in Figure 9. Calculated in this way, Table 4 shows the energy values and the energy per mass unit of the removed copper in experiments 1 and 5. The peaks of the two curves correspond to the point where the maximum voltage of the power supply is reached, and the current starts to drop, meaning a lower power.

According to Table 4, the energy consumption for 1.02 A was lower than the consumption for 1.36 A; for that reason, the operational costs of the second case will be higher. On the other hand, the residence time would be higher in the first case, which could lead to higher investment costs. Another parameter to consider is the ratio between energy consumption and the removed mass of copper. Table 4 shows that this energy consumption per mass of removed copper was similar for both cases. In accordance with the above, when a current of 1.36 A was used, shorter treatment times with the same efficiency of energy usage could be obtained compared to 1.02 A.

## 5. Conclusions

The use of combined electrodialysis (ED) and electrocoagulation (EC) to treat simulated copper smelter wastewater was proven successful using a batch reactor—achieving separation of copper from arsenic and simultaneous coagulation and precipitation of arsenic. This makes this process promising since copper can be recovered, and the final arsenic-containing residue has a lower volume than the waste generated in actual treatment plants. Furthermore, ferric iron addition for arsenic precipitation is avoided.

An important factor to consider is the applied current, since between 1.02 A and 1.36 A, it was observed that a decrease in operational time could be obtained associated with the increase in current maintaining a copper removal with a similar energy consumption per unit of mass, however at higher energy consumption. Both the ED and EC processes seem to be highly charge dependent.

The study of the behaviour of real wastewater and the application of this technique to recover other metal elements should be taken into account in future research, mainly due to the presence of other ions in the real wastewater. On the other hand, the oxidation of arsenic before adding it to Section 3 is recommended; this oxidation process would produce more mobile anions that could cross the anionic membrane and later be removed by EC.

## Figures and Tables

**Figure 1 membranes-13-00264-f001:**
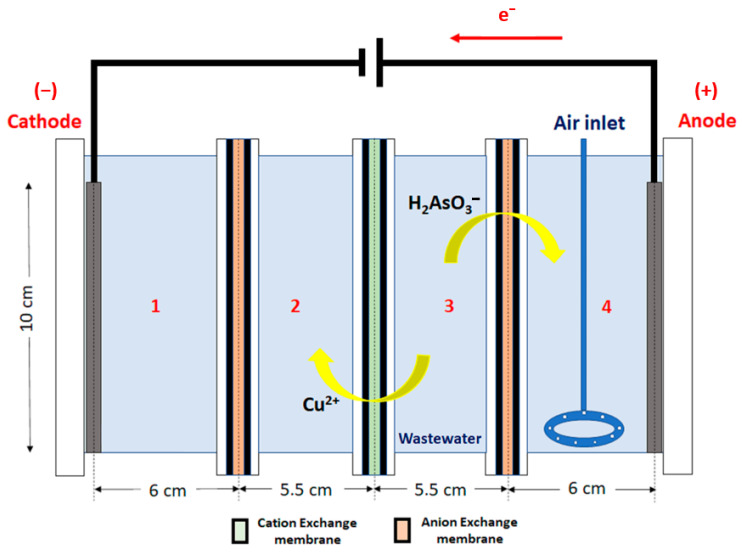
The experimental cell setup.

**Figure 2 membranes-13-00264-f002:**
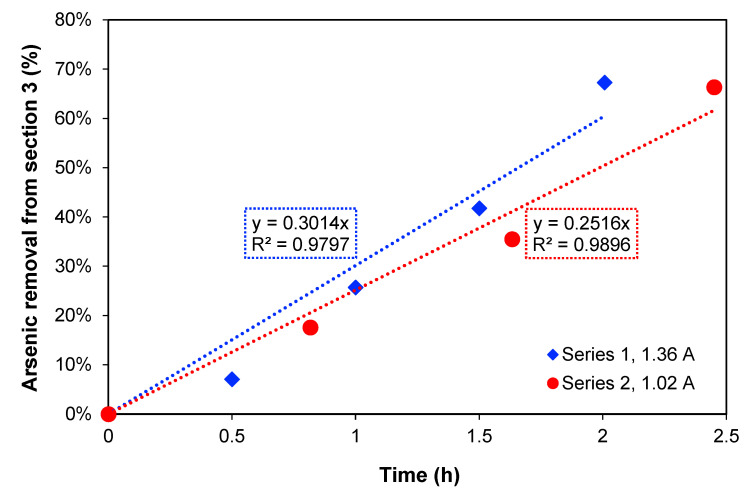
Percentage (%) of arsenic removal from Section 3 over time for the two series of currents.

**Figure 3 membranes-13-00264-f003:**
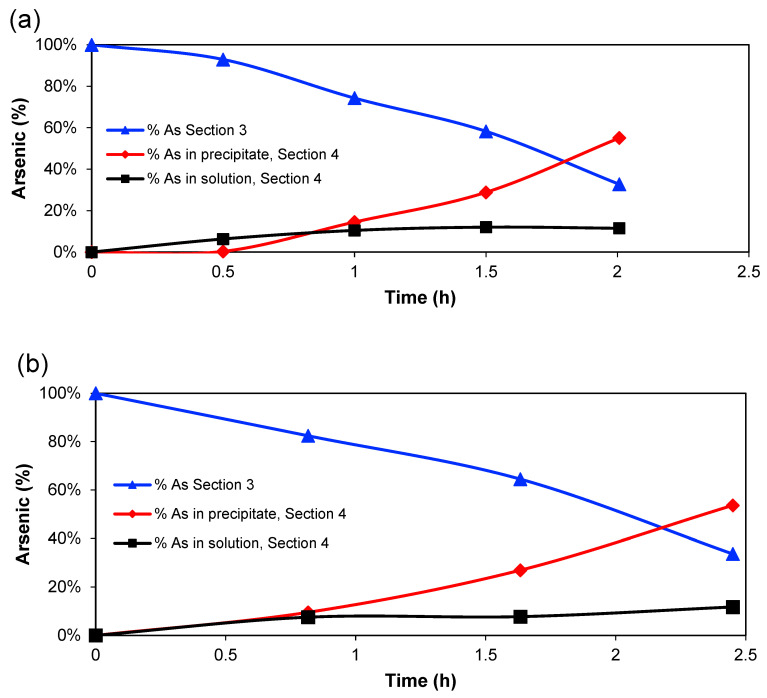
Percentage of arsenic in Section 3 and Section 4 over time (**a**) Series 1, (**b**) Series 2.

**Figure 4 membranes-13-00264-f004:**
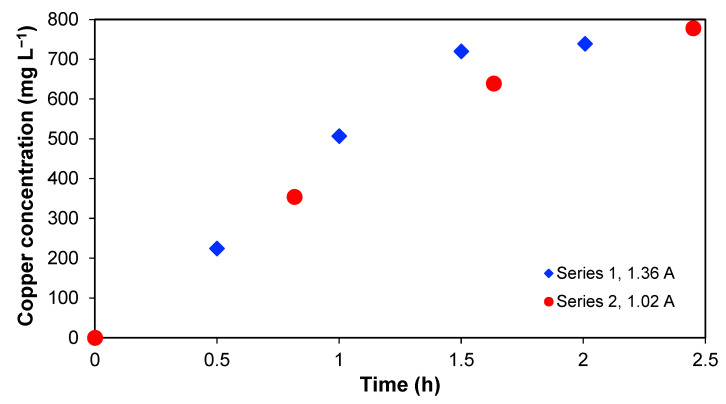
Copper concentration in Section 2 for series 1 and 2 as a function of time.

**Figure 5 membranes-13-00264-f005:**
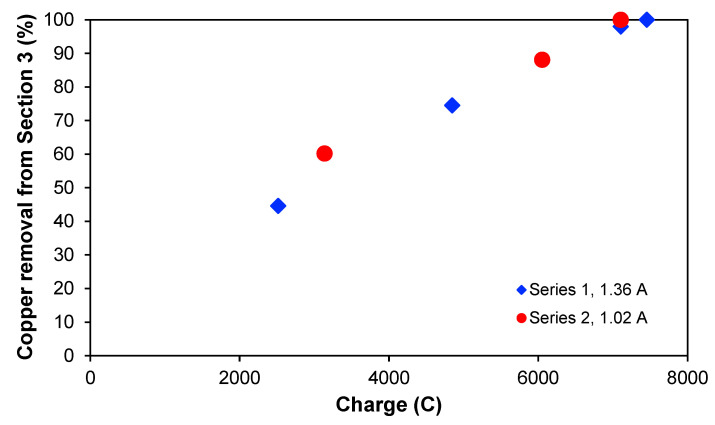
Copper removal (%) from Section 3 for series 1 and 2 as a function of passed electric charge.

**Figure 6 membranes-13-00264-f006:**
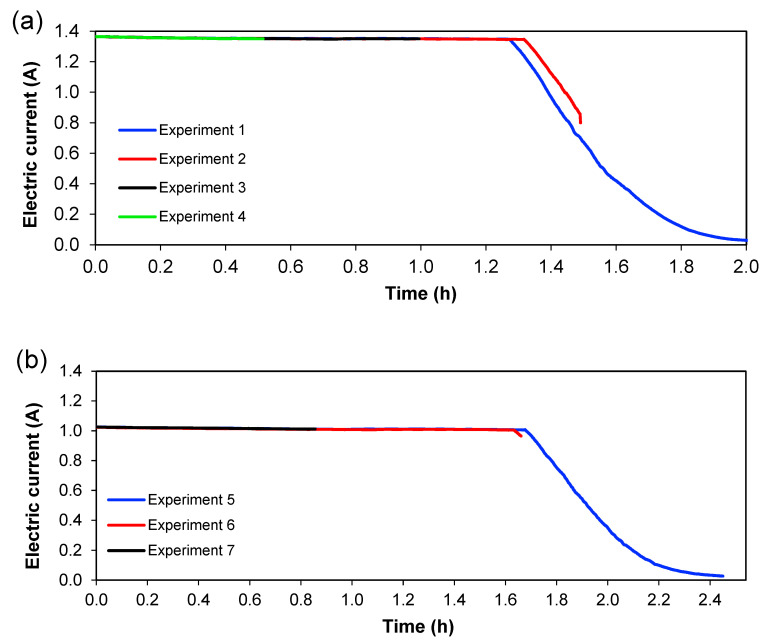
Development of the electric current with time. (**a**) Series 1, (**b**) Series 2.

**Figure 7 membranes-13-00264-f007:**
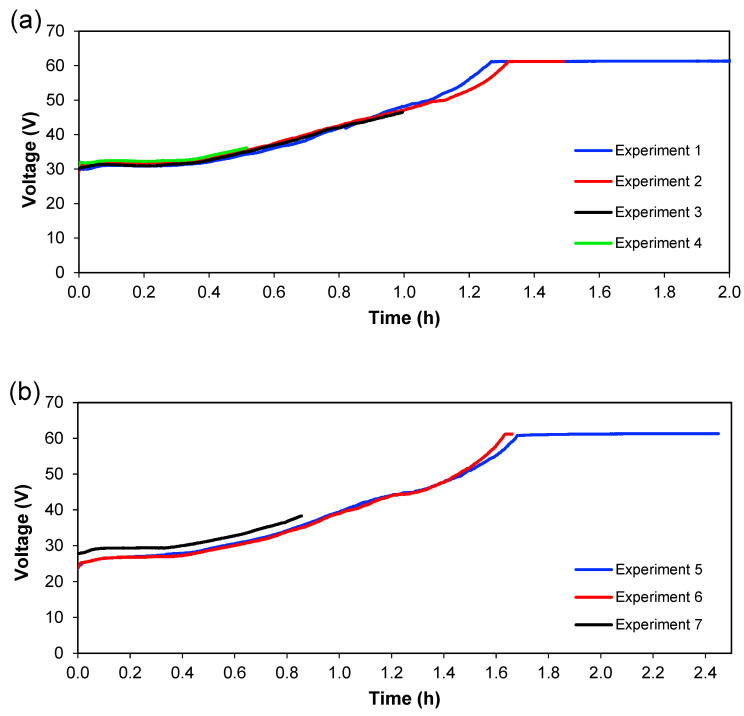
The voltage drop over the cell with time. (**a**) Series 1, (**b**) Series 2.

**Figure 8 membranes-13-00264-f008:**
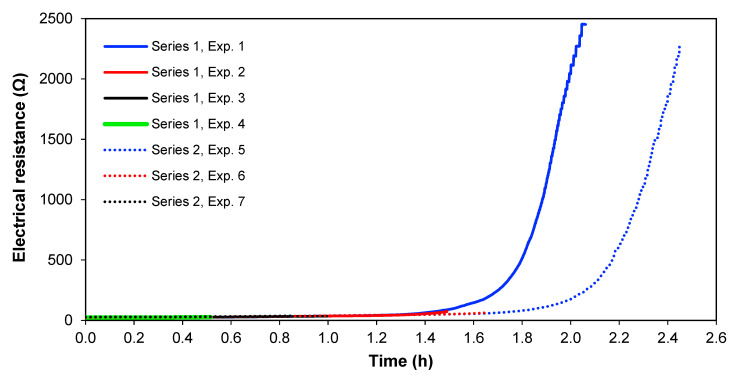
Electrical resistance across the cell with time.

**Figure 9 membranes-13-00264-f009:**
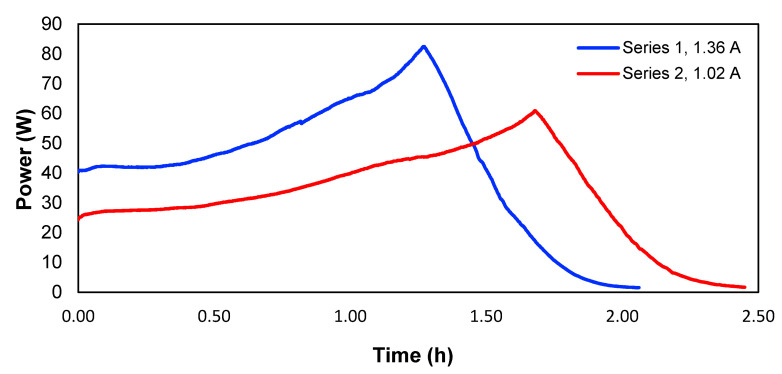
Power of the system over time.

**Table 1 membranes-13-00264-t001:** Experimental conditions and results.

Experimental Conditions														
Series	Exp.	Time h	Current, A	Current Density, A m^−2^		Section Volume, mL	Initial [Cu^2+^], mg L^−1^		Initial Cu^2+^ Mass, mg		Initial [As], mg L^−1^		Initial As Mass, mg	
1	1	2.01	1.36	194		450	814		366		7664		3449	
	2	1.50												
	3	1.00												
	4	0.50												
2	5	2.45	1.02	146		400	819		328		7794		3118	
	6	1.63												
	7	0.82												
**Results for Copper**														
**Series**	**Exp.**	** Section 2 **				** Section 3 **			** Section 4 **					
		**[Cu^2+^],** **mg L^−1^**	**Final Mass, mg**		$\;\%\;rec_{i.Cu,\;2}$	**[Cu^2+^],** **mg L^−1^**	**Final Mass, mg**	$\%\;rem_{i.Cu}$	**[Cu^2+^],** **mg L^−1^**	**Final Mass, mg**		$\;\%\;rec_{i.Cu,\;4}$		$\%Cu_{EC}$
1	1	739	333		91.0	<1	<1	100	<1	<1		<1		-
	2	720	324		88.5	16	7	98.0	<1	<1		<1		-
	3	507	228		62.3	208	93	74.5	<1	<1		<1		-
	4	224	101		27.6	451	203	44.6	<1	<1		<1		-
2	5	778	311		94.8	<1	<1	100	<1	<1		<1		-
	6	639	256		78.0	98	39	88.1	<1	<1		<1		-
	7	354	142		43.3	326	130	60.2	<1	<1		<1		-
**Results for Arsenic**														
**Series**	**Exp.**	** Section 2 **				** Section 3 **			** Section 4 **					
		**[As],** **mg L^−1^**	**Final Mass, mg**		$\;\%\;rec_{i.As,\;2}$	**[As],** **mg L^−1^**	**Final Mass, mg**	$\;\%\;rem_{i.As}$	**[As],** **mg L^−1^**	**Final Mass, mg**		$\;\%\;rec_{i.As,\;4}$		$\%As_{EC}\;$
1	1	57	25		<1	2507	1128	67.3	881	396		11.5		55.1
	2	67	30		<1	4464	2009	41.8	923	416		12.0		28.8
	3	57	25		<1	5694	2562	25.7	802	361		10.5		14.5
	4	30	13		<1	7122	3205	7.1	486	219		6.3		<1
2	5	76	31		<1	2622	1049	66.4	914	366		11.7		53.6
	6	67	27		<1	5026	2010	35.5	603	241		7.7		26.9
	7	41	16		<1	6424	2570	17.6	587	235		7.5		9.5

**Table 2 membranes-13-00264-t002:** Mass balance of arsenic in Section 4.

Exp	Time, h	Total Incoming Mass of As, mg	Mass of Coagulated Arsenic, mg	Percentage of Arsenic in Section 4 That is Coagulated, %
1	2.01	2296	1900	83
2	1.50	1410	994	71
3	1.00	862	501	58
4	0.50	231	12	5
5	2.45	2038	1672	82
6	1.63	1081	840	78
7	0.82	532	297	56

**Table 3 membranes-13-00264-t003:** Current efficiency with respect to copper and arsenic transport.

Experiment	Charge	Copper			Arsenic		
	C	M_theoi_, mg	M_expi_, mg	CE, %	M_theoi_, mg	M_expi_, mg	CE, %
1	7452	2454	365	14.9	5786	2296	39.7
2	7102	2339	359	15.3	5514	1410	25.5
3	4846	1596	273	17.1	3764	862	22.9
4	2515	828	163	19.7	1954	231	11.8
5	7103	2339	327	14.0	5516	2038	36.9
6	6051	1993	289	14.5	4698	1081	23.0
7	3136	1032	198	19.1	2456	532	21.8

**Table 4 membranes-13-00264-t004:** Consumed energy.

Experiment	Current, A	Energy, kJ	Energy/Copper Mass Removed, kJ/mg
1	1.36	360	0.98
5	1.02	320	0.97

## Data Availability

The data presented in this study are available on request from the corresponding author.

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
