# Peer review of "Combined Electrodialysis and Electrocoagulation as Treatment for Industrial Wastewater Containing Arsenic and Copper"

_membranes, 2023, doi:10.3390/membranes13030264_

Round 1

Reviewer 1 Report

In the work entitled with “Combined electrodialysis and electrocoagulation as treatment 2 for industrial wastewater containing arsenic and copper” electrodialysis (ED) and electrocoagulation (EC) were combined in an electrochemical reactor to separate some valuable cations from arsenic. I think the following points should be answered first before considering its publication in the journal.

Introduction and Theoretical background

It is poor. The work is about the development of a new reactor by the combination of ED and EC processes. Differently from ED process, the anodic reaction was used to oxidize arsenic in the system. The separation of anodic and cathodic reactions were very well studied in the literature and its is known that pH increases in the cathode side while it decreases in the anode part as a result of inhibition of OH- and H+ combination. The study separated anodic and cathodic reactions however the work was not mentioned about it.  Please check the Anderson and his co authors for the separated anodic and cathodic reactions and its effects on the oxidation and reduction reactions. Also please mention about advantages of increasing pH in the cathode and decreasing pH in the anode compartments. Also some disadvantages should be pointed out such as possible formation of carcinogenic by-product formation such as bromate in the anode part.

What is the aim of the combination of ED and EC processes? it is not clear. The advantages or disadvantages of this concept should be highlighted. What is the synergic effect of this combination? Or possible other effects? Economical? Low material usage? Low coagulant use? Possible other challenges? Membrane fouling due to the coagulation in the system?.............

Material and method

Figure 1 Transport of the Negatively and positively ions should be illustrated in the system please check some ED acid recovery works for that. Show arsenic movement to cathode and Cu+ transport to anode.

Why sulfuric acid is used in the cathode? To inhibit Cu+2 coagulation in cathode camper or Cu2+ fouling on membrane? Why not other electrolyte such as NaCl, Na2SO4 ……..?

What is the first solution in the section 2 to accumulate cupper?

Section 3: why low pH is needed to transport Cu+2 ions? Cu+2 is not mobile at high pH levels which has low H+ ions!? Or inhibit Cu+2 coagulation in the section 2?

Specification of cathode and anode exchange membrane should be given. Are they divalent ion exchange membranes?

It is written

“The synthetic wastewater was prepared by dissolving 3.158 g L-1 of CuSO4·5H2O and 178 14.047g L-1 of NaAsO2 (both analytical grade) in distilled water. 97.9% sulfuric acid (J.T. 179 Baker) ……………….” It is a synthetically wastewater? Or just a solution containing Cu2+ and AsO2. What is the real wastewater conditions? Real forms of As in the relevant condition?  As(III), As(V), cationic, anionic or both?  Alkalinity, other ions? 

Results

Current/voltage and electrical resistance changes were given without comments? Please check ED works (desalination and acid recovery works) to evaluate the changes of these parameters during treatment.

Author Response

Reviewer 1

In the work entitled with “Combined electrodialysis and electrocoagulation as treatment 2 for industrial wastewater containing arsenic and copper” electrodialysis (ED) and electrocoagulation (EC) were combined in an electrochemical reactor to separate some valuable cations from arsenic. I think the following points should be answered first before considering its publication in the journal.

 Introduction and Theoretical background

It is poor. The work is about the development of a new reactor by the combination of ED and EC processes. Differently from ED process, the anodic reaction was used to oxidize arsenic in the system. The separation of anodic and cathodic reactions were very well studied in the literature and its is known that pH increases in the cathode side while it decreases in the anode part as a result of inhibition of OH- and H+ combination. The study separated anodic and cathodic reactions however the work was not mentioned about it.  Please check the Anderson and his co authors for the separated anodic and cathodic reactions and its effects on the oxidation and reduction reactions. Also please mention about advantages of increasing pH in the cathode and decreasing pH in the anode compartments. Also some disadvantages should be pointed out such as possible formation of carcinogenic by-product formation such as bromate in the anode part.

Response: The idea and advantages of the proposed process have been stated more clearly. The main purpose with this process is to separate copper from arsenic (with a possible later recovery of copper) even if the concentration of arsenic is much higher. The secondary objective is to precipitate arsenic in the same cell using the same electric current as for the ED part to produce a much lesser volume of sludge with arsenic compared to conventional arsenic treating precipitation processes (adding reagents such as hydroxides or ferric salts). We do not have bromate or other bromine containing substances in the wastewater. Real copper smelter wastewater contains high amounts of sulfates, and this will migrate towards the anode competing with arsenic. The behavior of sulfate should be studied in future work. The referee should be more specific when writing “Please check the Anderson and his co authors for the separated anodic and cathodic reactions and its effects on the oxidation and reduction reactions.” There are several “Andersons” having references on applied electrochemistry, explaining effects of cathode and anode reactions. Furthermore, in our case it is not an advantage that pH increases at the cathode, because we want to keep pH low here to avoid precipitation of hydroxides here and fouling of the membranes. In addition, we cannot combine the cathode and anode solution since then we would mix copper and arsenic again.

What is the aim of the combination of ED and EC processes? it is not clear. The advantages or disadvantages of this concept should be highlighted. What is the synergic effect of this combination? Or possible other effects? Economical? Low material usage? Low coagulant use? Possible other challenges? Membrane fouling due to the coagulation in the system?.............

Response: Again, the main purpose with this process is to separate copper from arsenic (with a possible later recovery of copper) even if the concentration of arsenic is much higher. The secondary objective is to precipitate arsenic in the same cell using the same electric current as for the ED part to produce a much lesser volume of sludge with arsenic compared to conventional arsenic treating precipitation processes (adding reagents such as hydroxides or ferric salts). Therefore, overall, it would mean lower costs, less final residue (compared to conventional wastewater treatment methods), low reagent use among others.

Material and method

Figure 1 Transport of the Negatively and positively ions should be illustrated in the system please check some ED acid recovery works for that. Show arsenic movement to cathode and Cu+ transport to anode.

Response: The transport of Cu2+ and H2AsO3- has been indicated on Figure 1 in the revised manuscript.

Why sulfuric acid is used in the cathode? To inhibit Cu+2 coagulation in cathode camper or Cu2+ fouling on membrane? Why not other electrolyte such as NaCl, Na2SO4 ……..?

Response: Sulfuric acid is used to maintain a low pH in section 1 and to avoid precipitation of hydroxide and fouling of the membrane. In real copper smelter wastewater the content of sulfate and acidity is high, therefore we chose sulfuric acid to adjust pH to simulate the real wastewater better.

What is the first solution in the section 2 to accumulate cupper?

Response: The initial (first?) solution in section 2 was 0.5 M Sulfuric acid, and this was already indicated in the original manuscript.

Section 3: why low pH is needed to transport Cu+2 ions? Cu+2 is not mobile at high pH levels which has low H+ ions!? Or inhibit Cu+2 coagulation in the section 2?

Response: A low pH is not needed for the transport of Cu2+ but to avoid the precipitation of Cu(OH)2, which could occur since the anion exchange membrane between section 1 and 2 is not 100% charge selective - meaning that OH- produced at the cathode (Section 1) could enter section 2. Low pH (in Section 2) would neutralize the OH-.

Specification of cathode and anode exchange membrane should be given. Are they divalent ion exchange membranes?

Response: More specifications of the ion exchange membranes have been given.

It is written

“The synthetic wastewater was prepared by dissolving 3.158 g L-1 of CuSO4·5H2O and 178 14.047g L-1 of NaAsO2 (both analytical grade) in distilled water. 97.9% sulfuric acid (J.T. 179 Baker) ……………….” It is a synthetically wastewater? Or just a solution containing Cu2+ and AsO2. What is the real wastewater conditions? Real forms of As in the relevant condition?  As(III), As(V), cationic, anionic or both?  Alkalinity, other ions? 

Response: It is an aqueous solution that simulates a real wastewater in the best way to understand the behavior of the two elements of interest: copper and arsenic. Real copper smelter wastewater contain typically Cu2+ and both As(III) (>90% of total As – reference given in manuscript) and As(V) (<10%). In the first study, where the behavior of the process was investigated, for simplicity only As(III) species were chosen.

 Results

Current/voltage and electrical resistance changes were given without comments? Please check ED works (desalination and acid recovery works) to evaluate the changes of these parameters during treatment.

Response: Figures 6-8 show the current/voltage and electrical resistance changes but the adjacent text does comment and discuss the behavior. The aim of this work is not to desalinate seawater or brine nor the produce acid, so we don´t think the comparison to those processes are meaningful in this manuscript.

Reviewer 2 Report

Detailed comments:

1.      The English of the text should be checked

2.      Al line, please indicate what means f. ex. Also, more example must be indicated.

3.      At legends, indicated at figures, use the option superscript for unit of measure, correct A/dm2 with A/dm2

4.      At lines 289-291, authors write about current, where unit of measure is Adm-2. In the figure was indicated electrical current in A. Between current intensity and current density, exist a different, because current density is express in the literature by relation i = I/S, where i is the current density, I is current intensity, and S is the surface. So, correct in the manuscript.

Author Response

Reviewer 2

Detailed comments:

  1. The English of the text should be checked

Response: The English has been checked by an English speaking person.

  1. Al line, please indicate what means f. ex. Also, more example must be indicated.

Response: “F. ex.” stands for “for example”. We are not sure what the referee exactly means here but we have changed the text: ”…f. ex. recoverable copper” to “….recoverable copper and other elements such as arsenic, zinc and lead“.

  1. At legends, indicated at figures, use the option superscript for unit of measure, correct A/dm2 with A/dm2

Response: In the revised manuscript we use the current as measure, therefore the unit is A.

  1. At lines 289-291, authors write about current, where unit of measure is Adm-2. In the figure was indicated electrical current in A. Between current intensity and current density, exist a different, because current density is express in the literature by relation i = I/S, where i is the current density, I is current intensity, and S is the surface. So, correct in the manuscript.

Response: The reviewer is right. What we control and measure is the current (or current intensity). The current density is calculated from the dimensions of the cell, and therefore not an exact value. Therefore, current density (I/S) has been avoided or changed to current (I) in all text and figures.

Reviewer 3 Report

Summary and general comments

In this study, the authors combined electrodialysis and electrocoagulation methods to treat simulated wastewater containing arsenic and copper. The work is scientifically sound and the chosen methodology is suitable for this study. However, the introduction and the methodology sections can be improved. Please see the specific section below for detail:

Specific comments

1.   Abstract, I suggest authors to include the initial concentration of the copper and arsenic when mentioning the % removal.

2.   Introduction, there should be some discussion about the current development of ED and EC, and what authors intended to do, what are the electrode materials for EC,  what kind of ED membrane, and how it is different from current research

3.   The aims and objectives need to be mentioned clearly at the end of the introduction.

4.   There is no information whether the anode and cathode electrodes are purchased or made in the lab. Authors need to state these clearly. If there are purchased, authors should mention the manufacturer.

5.   Some errors such as “Cu+2” instead of “Cu2+”

6.   Line 184-186. What is the applied voltage?

7.   Fig 6(a) label should not be blocking the line of the graph

Author Response

Reviewer 3

Summary and general comments

In this study, the authors combined electrodialysis and electrocoagulation methods to treat simulated wastewater containing arsenic and copper. The work is scientifically sound and the chosen methodology is suitable for this study. However, the introduction and the methodology sections can be improved. Please see the specific section below for detail:

Specific comments

  1. Abstract, I suggest authors to include the initial concentration of the copper and arsenic when mentioning the % removal.

Response: The initial concentrations have been added.

  1. Introduction, there should be some discussion about the current development of ED and EC, and what authors intended to do, what are the electrode materials for EC,  what kind of ED membrane, and how it is different from current research.

Response: In the revised manuscript, more recent trends in ED and EC research have been added including more references. Furthermore, the idea and advantages of the proposed process have be stated and discussed more clearly. In Experimental section more details about the membranes used have been given, The EC electrode materials were carbon steel (anode) and stainless steel (cathode) – already stated in the original manuscript.

  1. The aims and objectives need to be mentioned clearly at the end of the introduction.

Response: The aims and objective have been indicated more clearly in the revised manuscript.

  1. There is no information whether the anode and cathode electrodes are purchased or made in the lab. Authors need to state these clearly. If there are purchased, authors should mention the manufacturer.

Response: The electrodes were homemade in the lab.

  1. Some errors such as “Cu+2” instead of “Cu2+”

Response: Error corrected.

  1. Line 184-186. What is the applied voltage?

Response: We didn’t apply a fixed voltage. The current was fixed and the voltage was monitored, which can be observed in Figure 7.

  1. Fig 6(a) label should not be blocking the line of the graph

Response: Labels a) and b) have been moved out of the graph area on all figures.

Round 2

Reviewer 3 Report

After the revision, the quality of the manuscript improved. Previously raised issues/ errors were clarified/ corrected. I recommend this manuscript for publication in this journal.

Author Response

Reviewer: "After the revision, the quality of the manuscript improved. Previously raised issues/ errors were clarified/ corrected. I recommend this manuscript for publication in this journal."

Response: "Thank you very much for your recommendation"